# Rapid screening for severe acute respiratory syndrome coronavirus 2 infection with a combined point-of-care antigen test and an immunoglobulin G antibody test

**Kosuke Mori** [1]*, **Shohei Imaki**[1], **Yutaro Ohyama**[1], **Kosuke Satoh**[1], **Takeru Abe**[2], **Ichiro Takeuchi**[2]

**1** Yokohama Municipal Citizen's Hospital, Yokohama, Japan, **2** Yokohama City University Medical Center, Yokohama, Japan

* morimati81@yahoo.co.jp

## Abstract

Rapid screening and diagnosis of coronavirus disease 2019 in the emergency department is important for controlling infections. When polymerase chain reaction tests cannot be rapidly performed, rapid antigen testing is often used, albeit with insufficient sensitivity. Therefore, we evaluated the diagnostic accuracy of combining rapid antigen and antibody test results. This was a retrospective review of patients who visited our emergency department between February and May 2021 and underwent rapid antigen, immunoglobulin G antibody, and reverse transcription–polymerase chain reaction tests. The study included 1,070 patients, of whom 56 (5.2%) tested positive on reverse transcription–polymerase chain reaction. The sensitivity, specificity, and area under the curve of rapid antigen testing were 73.7%, 100.0%, and 0.87, respectively. The combined rapid antigen and antibody test result had improved diagnostic accuracy, with 91.2% sensitivity, 97.9% specificity, and an area under the curve of 0.95. The results of the rapid antigen and antibody tests could be combined as a reliable alternative to reverse transcription–polymerase chain reaction.

## Introduction

The coronavirus disease 2019 (COVID-19) pandemic is a major worldwide problem [1]. There are three main routes of virus transmission [2]: (i) inhalation of very fine respiratory droplets and aerosol particles [2–5]; (ii) deposition of respiratory droplets and particles on exposed mucous membranes in the mouth, nose, or eyes; and (iii) contact between mucous membranes and hands soiled either directly by virus-containing respiratory fluids or indirectly by touching surfaces contaminated with the virus. Of note, it is still controversial whether severe acute respiratory syndrome coronavirus 2 (SARS-CoV-2) causes airborne infection and whether N95 masks are superior to surgical masks for protection against infection [6]. However, even if hospitals do not have sufficient amounts of personal protective equipment, we must provide medical care to patients with COVID-19. This poses a risk of nosocomial

**Data Availability Statement:** All relevant data are within the paper and its Supporting information files.

**Funding:** The author(s) received no specific funding for this work.

**Competing interests:** The authors have declared that no competing interests exist.

infections among medical personnel [7, 8]. Therefore, early screening for COVID-19 in patients who visit the emergency department (ED) is important.

The gold standard for diagnosis of COVID-19 is the detection of severe acute respiratory syndrome coronavirus 2 (SARS-CoV-2) RNA using nucleic acid amplification tests (NAATs) with specimens obtained from the upper respiratory tract [9]. NAATs are divided into two categories: laboratory-based NAATs mainly comprising reverse transcription–polymerase chain reaction (RT-PCR), and point-of-care (POC) NAATs that are isothermal amplifications, such as nicking endonuclease amplification reaction and loop-mediated isothermal amplification [9]. Although RT-PCR is the most reliable and widely used NAAT, there are concerns regarding false-negative results. Long *et al.* [10] reported that the false-negative rate of RT-PCR was approximately 3.5%. However, caution should be exercised while interpreting the estimated rate because there is no gold standard for testing to compare with. The disadvantages of laboratory-based NAATs include long turnaround times and limited number of tests and locations where tests can be performed. Rapid antigen testing is often used for screening when laboratory-based NAATs cannot be performed rapidly. However, its sensitivity is low (68.9%; 95% confidence interval [CI]: 61.8–75.1) [11], and its diagnostic ability decreases 7 days after onset of disease, with sensitivities of 78.3% (95% CI: 71.1–84.1) and 51.0% (95% CI: 40.8–61.0) in the first and second week, respectively [11]. Therefore, even if these rapid antigen test results were negative, the patients may still have been infected. Some POC NAATs are sometimes used as substitutes for laboratory-based NAATs. However, they are not very accurate. The sensitivity of Abbott ID NOW (Abbott, Chicago, IL, USA), a POC NAAT using nicking endonuclease amplification reaction, was reported to be 81% [11]. However, the number of facilities that can perform this test is limited.

Since the sensitivity of antibody testing increases with time after disease onset, it has the potential to compensate for the limitations of rapid antigen testing. In one study, the sensitivity within the first week of illness was low. However, after the third week of illness, the sensitivity of immunoglobulin G (IgG) was 80.3% (95% CI: 72.4–86.4); immunoglobulin M (IgM), 68.1% (95% CI: 55.0–78.9); IgG/IgM, 96.0% (95% CI: 90.6–98.3) (either IgG- or IgM-positive indicating a positive result); and immunoglobulin A (IgA), 98.7% (95% CI: 91.9–99.8) [12].

Thus, rapid antigen tests are useful when performed early in the course of the disease, while antibody testing is useful three weeks after disease onset. In this study, we evaluated the diagnostic accuracy of a combination of rapid antigen and antibody tests for diagnosis of COVID-19 in the acute phase, as this has not been sufficiently verified.

## Materials and methods

### Study population

All patients who visited the ED between February 1, 2021 and May 31, 2021 and underwent rapid antigen and IgG antibody tests and RT-PCR for diagnosis or screening for COVID-19, regardless of the reason for the visit, were included in the study. No specific exclusion criteria were determined.

### Study design

This was a single-center retrospective medical chart review. All procedures performed in studies involving human participants were in accordance with the ethical standards of the institutional and/or national research committee and with the 1964 Declaration of Helsinki and its later amendments or comparable ethical standards. The study design was approved by the Institutional Review Board and the Ethics Committee of Yokohama Municipal Citizen's Hospital (approval number: 210602). Our hospital is an emergency medicine center located

centrally in Yokohama City, which has a population of 3.78 million. Between April 2020 and March 2021, the total number of emergency patients was 12,078, the number of ambulance transports was 4,710, and the number of patients admitted via the emergency room was 4,853 [13]. The total number of patients diagnosed with COVID-19 in the city during the study period was 8,469 [14].

In principle, nasopharyngeal swab specimens for rapid antigen testing and RT-PCR were collected on same-day ED visit. The samples were analyzed using SARS-CoV-2 rapid antigen tests (Roche Diagnostics, Basel, Switzerland), the Anility SARS-CoV-2 IgG assay (Abbott), and the Cobas z480 PCR analyzer (Roche Diagnostics), respectively. The reason, for choosing these antigen and antibody tests was simply that they could be performed in our hospital. Rapid antigen tests use immunochromatography to qualitatively detect SARS-CoV-2 antigens. Antibody assays are chemiluminescent microparticle immunoassay for semi-quantitative detection of IgG in human serum or plasma against SARS-CoV-2 nucleoproteins. Results are expressed as a ratio (S/C) of the luminescence intensity of patient's sample to that of a control. The cutoff value was 1.4 (S/C). If either the rapid antigen or the IgG antibody test was positive, the sample was considered positive. A positive RT-PCR test was considered the gold standard for the diagnosis of COVID-19. A negative RT-PCR test, despite antibody positivity, was considered as a previous infection.

## Statistical analysis

Data on demographic and clinical characteristics of the patients were collected, including age (years), sex (male or female), clinical symptoms (respiratory, fever, and/or diarrhea), and the number of days from disease onset to hospital visit. Sensitivity, specificity, positive predictive value (PPV), negative predictive value (NPV), area under the curve (AUC), and 95% CI of rapid antigen test, IgG antibody test, and a combination of both were calculated. Regarding IgG antibody testing, sensitivity analysis was performed using cutoff values of 0.1 (S/C), 0.5 (S/C), and 1.0 (S/C), in addition to the manufacturer's recommended cutoff value of 1.4 (S/C). We chose 0.1 (S/C) as one cutoff value because we hypothesized that, based on our experience, many non-COVID-19 patients would have antibody levels below 0.1. Other potential cutoff values were selected between 0.1 and 1.4. Among patients with positive RT-PCR results, the number of days after symptom onset was calculated considering rapid antigen and antibody test results. We obtained median (interquartile range, IQR) for continuous variables and frequency (%) for categorical variables. In addition, we used the Wilcoxon rank sum test to compare between the two groups by antigen and antibody results. Moreover, data on the clinical characteristics of patients with positive RT-PCR results but negative rapid antigen and antibody test results were obtained. Statistical significance was set at $p < 0.05$. Statistical analysis was performed using Stata Statistical Software: Release 12.1. (StataCorp, College Station, TX, USA).

## Results

### Patient characteristics

A total of 1,070 patients visited our ED and underwent all antigen, antibody, and RT-PCR tests. The total number of RT-PCR-positive patients was 56 (5.2%). The median age of the RT-PCR-positive patients was 77 years (IQR: 56–86 years), and 23 (41.1%) were men. The characteristics of the RT-PCR-positive patients are shown in Table 1. Among the RT-PCR-positive patients, 94.6% had respiratory symptoms (sore throat, cough, sputum, and nasal discharge) and 58.9% visited our ED within a week of symptom onset.

**Table 1. Characteristics of patients with COVID-19.**

| | Patients with COVID-19, No. (%)[a] n = 56 |
|---|---|
| Age (median, IQR) | 77 (56–86) |
| Males | 23 (41.1) |
| Clinical symptoms | |
| Respiratory symptoms | 53 (94.6) |
| Fever | 42 (75.0) |
| Diarrhea | 1 (1.8) |
| Number of days from onset | |
| Median (IQR) | 5 (3–7) |
| Day 1–7 | 33 (58.9) |
| Day 8–14 | 19 (33.9) |
| Day 15–21 | 2 (3.6) |
| Day 22– | 2 (3.6) |

COVID-19, coronavirus disease 2019; IQR: interquartile range.

[a]Data are presented as no. (%) of participants unless otherwise specified.

## Rapid antigen test and IgG antibody assay

The results of rapid antigen and antibody testing are shown in Table 2. Of the 1,070 individuals who underwent all three tests, rapid antigen tests were positive in 42 patients (3.9%) and were found to have a sensitivity of 73.7% (95% CI: 60.3–84.5%), specificity of 100.0% (95% CI: 99.6–100.0%), PPV of 100.0% (95% CI: 91.6–100.0%), NPV of 98.5% (95% CI: 97.6–99.2%), and an AUC of 0.87 (95% CI: 0.85–0.89). IgG antibody assays were positive in 35 patients (3.3%) and had sensitivity of 24.6% (95% CI: 14.1–37.8%), specificity of 97.9% (95% CI: 96.8–98.7%), PPV of 40% (95% CI: 23.9–57.9%), NPV of 95.8% (95% CI: 94.4–97.0%), and an AUC of 0.61 (95% CI: 0.58–0.64).

## Combination of rapid antigen and IgG antibody tests

The combination of rapid antigen and IgG antibody tests was positive in 73 patients (6.8%; 95% CI: 5.3–8.3%), with a sensitivity of 91.2% (95% CI: 80.7–97.1%), specificity of 98.0% (95% CI: 96.8–98.7%), PPV of 71.2% (95% CI: 59.4–81.2%), NPV of 99.5% (95% CI: 98.8–99.8%), and AUC of 0.95 (95% CI: 0.93–0.96). The results are shown in Table 3. Regarding the RT-PCR-positive patients, only four patients tested negative on both rapid antigen and antibody testing.

## Sensitivity analysis of cutoff values for IgG antibody assays

The analysis was performed at four cutoff indexes: 0.1, 0.5, 1.0, and 1.4 (S/C). The diagnostic accuracies at each cutoff index for the combined IgG antibody assay and rapid antigen test are shown in Table 4. Comparing the four cutoff values, the sensitivity was slightly better with the

**Table 2. Diagnostic accuracy of the rapid antigen test and IgG antibody assay.**

| | Sensitivity % (95% CI) | Specificity % (95% CI) | PPV % (95% CI) | NPV % (95% CI) | AUC (95% CI) |
|---|---|---|---|---|---|
| Antigen test | 73.7 (60.3–84.5) | 100 (99.6–100.0) | 100 (91.6–100.0) | 98.5 (97.6–99.2) | 0.87 (0.85–0.89) |
| IgG antibody assay | 24.6 (14.1–37.8) | 97.9 (96.8–98.7) | 40 (23.9–57.9) | 95.8 (94.4–97.0) | 0.61 (0.58–0.64) |

AUC, area under the curve; CI, confidence interval; IgG, immunoglobulin G; NPV, negative predictive value; PPV, positive predictive value.

**Table 3. Diagnostic accuracy of the combination of rapid antigen test and IgG antibody assay.**

|  | Sensitivity % (95% CI) | Specificity % (95% CI) | PPV % (95% CI) | NPV % (95% CI) | AUC (95% CI) |
|---|---|---|---|---|---|
| Antigen, IgG | 91.2 (80.7–97.1) | 97.9 (96.8–98.7) | 71.2 (59.4–81.2) | 99.5 (98.8–99.8) | 0.95 (0.93–0.96) |

AUC, area under the curve; CI, confidence interval; IgG, immunoglobulin G; NPV, negative predictive value; PPV, positive predictive value.

cutoff value of 0.1, but the NPVs were almost similar. However, the PPV increased as the cutoff values increased, from 33.3% to 57.1%, 68.4%, and 71.2%. In addition, the AUC of the cutoff value of 1.4 was slightly higher than the AUCs of the other cutoff values.

## Comparison of the number of days after symptom onset for rapid antigen and IgG antibody testing

The number of days from symptom onset was compared between rapid antigen-positive and -negative patients. The median (IQR) of the number of days from symptom onset was 5 (3–7) days in rapid antigen-positive patients and 12 (7–18) days in rapid antigen-negative patients, showing a statistically significant difference ($p = 0.0009$). Similarly, IgG antibody test results showed the number of days from symptom onset to be 12 (7–18) days in IgG antibody-positive patients and 4 (2–7) days in IgG antibody-negative patients, also with a statistically significant difference ($p < 0.0001$). In addition, the minimum number of days from disease onset in patients who were negative for rapid antigen tests but positive for IgG antibody and RT-PCR tests was 7 days.

## Characteristics of four patients with COVID-19 with negative results for both rapid antigen and IgG antibody assays

Four patients (0.4%) had positive RT-PCR test results despite negative results of both rapid antigen test and IgG antibody tests. These patients had a history of close contact with patients with COVID-19 who were family members, coresidents, or caregivers. In addition, all patients had fever, respiratory symptoms, and other clinical symptoms suggestive of COVID-19. The characteristics of these patients are presented in Table 5.

## Discussion

In this study, we examined the diagnostic accuracy of a combination of rapid antigen and IgG antibody tests using RT-PCR as the gold standard. The sensitivity, specificity, PPV, NPV, and AUC of rapid antigen tests were higher than those of IgG antibody tests. There are few reports on the use of antibody tests for diagnosis in the acute phase of the disease [12]. Moreover, the accuracy of this antibody test used in the present study was not sufficient for it to be the sole

**Table 4. Sensitivity analysis of the test combination with various cutoff values for the IgG antibody assay.**

|  | Sensitivity % (95% CI) | Specificity % (95% CI) | PPV % (95% CI) | NPV % (95% CI) | AUC (95% CI) |
|---|---|---|---|---|---|
| Antigen, IgG > 1.4 (S/C) | 91.2 (80.7–97.1) | 97.9 (96.8–98.7) | 71.2 (59.4–81.2) | 99.5 (98.8–99.8) | 0.95 (0.93–0.96) |
| Antigen, IgG ≥ 1.0 (S/C) | 91.2 (80.7–97.1) | 97.6 (96.5–98.5) | 68.4 (56.7–78.6) | 99.5 (98.8–99.8) | 0.94 (0.93–0.96) |
| Antigen, IgG ≥ 0.5 (S/C) | 91.2 (80.7–97.1) | 96.2 (94.8–97.2) | 57.1 (46.3–67.5) | 99.5 (98.8–99.8) | 0.94 (0.92–0.95) |
| Antigen, IgG ≥ 0.1 (S/C) | 93.0 (83.0–98.1) | 89.5 (87.5–91.4) | 33.3 (26.1–41.2) | 99.6 (98.9–99.9) | 0.91 (0.89–0.93) |

AUC, area under the curve; CI, confidence interval; IgG, immunoglobulin G; NPV, negative predictive value; PPV, positive predictive value.

**Table 5. Characteristics of four patients with COVID-19 with negative antigen and antibody results.**

| No. | Age (y) | Sex | Respiratory symptoms | Fever | Close contact with a COVID-19 patient | Number of days from the onset of symptoms to ED |
|---|---|---|---|---|---|---|
| 1 | 86 | M | + | + | + | 4 |
| 2 | 92 | F | + | − | + | 1 |
| 3 | 87 | M | + | + | + | 3 |
| 4 | 78 | M | + | + | + | 8 |

COVID-19, coronavirus disease 2019; ED, emergency department.

diagnostic tool. However, the diagnostic accuracy greatly improved when rapid antigen testing was combined with IgG antibody testing.

A Cochrane systematic review and meta-analysis in 2021 reported a sensitivity of 68.9% (95% CI: 61.8–75.1) and a specificity of 99.6% (95% CI: 99.0–99.8) for rapid antigen testing [11]. The sensitivity was 78.3% (95% CI: 71.1–84.1) when used during the first 7 days of symptoms. However, this sensitivity decreased to 51.0% (95% CI: 40.8–61.0) during the second week of symptoms [11]. Considering that approximately half the patients in this study had symptom onset within 7 days, and the median number of days since symptom onset in positive patients was 5 days. The sensitivity of rapid antigen testing (73.7%) in this study is consistent with the results of previous studies. Antibody testing was also studied in 2020 [12]. Furthermore, the sensitivities of IgG were found to be 29.7% (95% CI: 22.1–38.6), 66.5% (95% CI: 57.9–74.2), and 88.2% (95% CI: 83.5–91.8) in the first, second, and third week, respectively. This suggests that sensitivity increased with time from disease onset; thus, antibody testing is not a useful diagnostic tool during the early stages of the disease. Similarly, the sensitivities of IgM were 23.2% (95% CI: 14.9–34.2), 58.4% (95% CI: 45.5–70.3), and 75.4% (95% CI: 64.3–83.8) during the first, second, and third weeks, respectively. In addition, the sensitivities of IgA were 28.4% (95% CI: 0.9–94.3), 78.1% (95% CI: 9.5–99.2), and 98.7% (95% CI: 39.0–100) during the same three weeks, respectively. Furthermore, the sensitivities of combination of IgG/IgM were 30.1% (95% CI: 21.4–40.7), 72.2% (95% CI: 63.5–79.5), and 91.4% (95% CI: 87.0–94.4) during these same three weeks, respectively. Therefore, the sensitivity of all antibody levels was low within the first week of disease onset [12], and it is unclear whether IgG, IgM, IgA, and IgG/IgM antibodies are preferable for diagnosing acute infection. After the third week, the sensitivity of IgG was 80.3% (95% CI: 72.4–86.4) and of IgM was 68.1% (95% CI: 55.0–78.9). This indicated that if either IgG or IgM was positive, its sensitivity was 96.0% (95% CI: 90.6–98.3); the sensitivity of positive IgA was 98.7% (95% CI: 91.9–99.8) [12]. Moreover, the specificity of all antibodies was > 98% throughout the study period.

These results indicate that rapid antigen testing is useful early in the course of the disease, especially in the first week, and antibody testing is useful later, especially after the third week. In situations where RT-PCR cannot be performed rapidly, a combination of antigen and antibody tests can be considered as a substitute for RT-PCR. However, in cases where pretest probability is high, caution should be exercised until acute SARS-CoV-2 infection is ruled out, by including RT-PCR test results. In our study, four patients with positive RT-PCR results tested negative on both rapid antigen and antibody testing. However, all four patients had a clear history of close contact with COVID-19-confirmed patients, had clinical symptoms strongly suggestive of COVID-19, and were considered clinically to be COVID-19-positive, regardless of their antigen and antibody test results.

This study has several limitations. First, it did not include all patients who visited the ED. Only patients who visited our ED and underwent all three tests, namely rapid antigen,

antibody, and RT-PCR, were included. Although an NPV of 99.5% suggests that our results are robust, the accuracy of this study, as it pertains to all patients who visited the ED, is unknown. Second, the patients were not grouped according to presence or absence of symptoms. The accuracy of the tests may have varied depending on presence or absence of suspected COVID-19 symptoms, such as fever and respiratory symptoms [11]. However, in this study, we were not able to obtain information on the presence of respiratory symptoms in all patients; therefore, we were unable to evaluate this issue. Third, it is unclear which type of antibody test is best for screening acute SARS-CoV-2 infection. From the viewpoint of rapid screening, qualitative tests are better than quantitative tests. Moreover, it is not clear which serotypes and detection methods are desirable. In our study, we used quantitative antigen tests and qualitative antibody tests; therefore, it took approximately one hour to obtain antibody results; this is a drawback for rapid screening. IgG or total antibodies and the enzyme-linked immunosorbent assay are recommended for screening early and late SARS-CoV-2 infections [15–18]. Fourth, a single negative RT-PCR result is the gold standard for diagnosis. However, in some patients, a single negative RT-PCR result may have not ruled out COVID-19 [10], and another RT-PCR test may have been necessary.

## Conclusion

A combination of rapid antigen and antibody tests could be an alternative to RT-PCR testing. However, caution should be exercised when pretest probability is high, such as when the prevalence in the area is extremely high or there is a history of close contact with COVID-19 patients. Further studies are required to confirm this conclusion.

## Supporting information

**S1 Data.**
(XLSX)

## Acknowledgments

We would like to express our heartfelt gratitude to Yuichi Honma and all members of the laboratory of Yokohama Municipal Citizen's Hospital and thank Editage (www.editage.com) for their writing support.

## Author Contributions

**Conceptualization:** Kosuke Mori, Takeru Abe, Ichiro Takeuchi.

**Data curation:** Kosuke Mori, Yutaro Ohyama, Kosuke Satoh.

**Formal analysis:** Kosuke Mori, Takeru Abe.

**Investigation:** Kosuke Mori, Yutaro Ohyama, Kosuke Satoh.

**Methodology:** Kosuke Mori, Takeru Abe.

**Project administration:** Kosuke Mori.

**Supervision:** Shohei Imaki, Ichiro Takeuchi.

**Validation:** Takeru Abe.

**Visualization:** Kosuke Mori.

**Writing – original draft:** Kosuke Mori.

**Writing – review & editing:** Shohei Imaki, Takeru Abe, Ichiro Takeuchi.

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
