## [Decision Letter · Decision Letter 0]

10 Dec 2021

PONE-D-21-33926Rapid screening for severe acute respiratory syndrome coronavirus 2 infection with a combined point-of-care antigen test and an immunoglobulin G antibody testPLOS ONE

Dear Dr. Mori, 

Thank you for submitting your manuscript to PLOS ONE. After careful consideration, we feel that it has merit but does not fully meet PLOS ONE’s publication criteria as it currently stands. Therefore, we invite you to submit a revised version of the manuscript that addresses the points raised during the review process. Both reviewers have commented on your statistical analysis. You need to explain your data and also to provide a justification for the used test in each graph. Maybe you can do that in the Legend to Figures. 

We look forward to receiving your revised manuscript.

Kind regards,

Gheyath K. Nasrallah

Academic Editor

PLOS ONE

Journal Requirements:

Reviewers' comments:

Reviewer's Responses to Questions

**Comments to the Author**

1. Is the manuscript technically sound, and do the data support the conclusions?

Reviewer #1: Yes

Reviewer #2: Yes

2. Has the statistical analysis been performed appropriately and rigorously? 

Reviewer #1: Yes

Reviewer #2: Yes

3. Have the authors made all data underlying the findings in their manuscript fully available?

Reviewer #1: Yes

Reviewer #2: Yes

4. Is the manuscript presented in an intelligible fashion and written in standard English?

Reviewer #1: Yes

Reviewer #2: Yes

5. Review Comments to the Author

Reviewer #1: In the study design section:

1- You mentioned the history of admitted patients from April 2019 – March 2020. However, the study period is a year later. Updating this information to include the studied period would help in better reflecting the prevalence of COVID19 positive cases.

2- Please add why you chose these two particular tests and more details, example; detect antibodies against which protein and the detection limit.

In the statistical analysis section:

3- You mentioned that you performed the analysis using other cutoff values then identified by the kits, please elaborate why these particular values were selected.

In the result section:

4- Line115: Add the total number and percentage of COVID-19 cases in text.

5- Line 126: Mention that percentage was calculated from total number of tested cases.

6- Line 141 you stated that only 4 tested positive on both rapid tests which doesn’t align with your data. In Line 169-175 as well as in line 210-211 you mentioned that these four cases were negative. Please correct the information.

In the discussion section:

7- Line 199-200; you could include more details about the synthesis time of each of the antibodies.

8- Line 233: state the actual accuracy of RT-PCR in this context.

In the references section

9- There are two set of references! Please revise and combine them.

Reviewer #2: The manuscript described the performance of combined rapid antigen and antibody test. The results showed an improved diagnostic accuracy, with 92.9% sensitivity, 98.0% specificity. These data can be useful to clarify the role and the potential utility of the combination of rapid antigen and antibody tests that could be an alternative to RT-PCR testing. The manuscript is well written.

However, the paper needs some minor revision before being published.

The authors should indicate which statistical test used when they made the comparison of the number of days after symptom onset for rapid antigen testing and IgG antibody testing. The authors report only the significances with a p equal to 0.0001. They also report the mean and standard deviation values, where sometimes the standard deviation is higher than the mean. Perhaps they need to check the distribution of the data and choose the most appropriate statistical test.

6. PLOS authors have the option to publish the peer review history of their article (what does this mean?). If published, this will include your full peer review and any attached files.

Reviewer #1: **Yes: **Fatiha Benslimane

Reviewer #2: No

---

## [Author Response · Author response to Decision Letter 0]

31 Dec 2021

Our point-by-point responses to all comments and suggestions are listed below:

TO THE EDITOR and REVIEWERS:

We thank you for your thorough review and insightful comments on our manuscript. We numbered your comments for convenience. Please find our responses to each comment below. Although we believe that we have addressed all of the reviewers’ comments, we would be more than pleased to answer any further query you might have.

In addition, we apologize for some unintentional errors in the data; we have revised the results in the manuscript and tables accordingly.

Reviewer #1

 In the study design section:

1-1 You mentioned the history of admitted patients from April 2019 – March 2020. However, the study period is a year later. Updating this information to include the studied period would help in better reflecting the prevalence of COVID19 positive cases.

RESPONSE 1-1

We thank you for your comment. Accordingly, the data you pointed out has been replaced with the data from April 2020 to March 2021. 

Pages: 5, Lines 86-89

Between April 2020 and March 2021, the total number of emergency patients was 12,078, the number of ambulance transports was 4,710, and the number of patients admitted via the emergency room was 4,853 [13].

1-2 Please add why you chose these two particular tests and more details, example; detect antibodies against which protein and the detection limit.

RESPONSE 1-2

We thank you for your comment. The reason, for choosing these antigen and antibody tests was simply that they could be performed in our hospital. We have added an explanation of antigen and antibody testing to the text.

Page: 6, Lines: 94-99

 The reason, for choosing these antigen and antibody tests was simply that they could be performed in our hospital. Rapid antigen tests use immunochromatography to qualitatively detect SARS-CoV-2 antigens. Antibody assays are chemiluminescent microparticle immunoassay for semi-quantitative detection of IgG in human serum or plasma against SARS-CoV-2 nucleoproteins. Results are expressed as a ratio (S/C) of the luminescence intensity of patient's sample to that of a control.

In the statistical analysis section:

1-3 You mentioned that you performed the analysis using other cutoff values then identified by the kits, please elaborate why these particular values were selected.

RESONSE 1-3

We thank you for your comment and apologize for this lack of explanation. We chose 0.1 as one cutoff value because we believed that, based on our experience, many non-COVID-19 patients would have antibody levels below 0.1. Other potential cutoff values were selected between 0.1 and 1.4. We added an explanation of the selected cutoff values to the statistical analysis section of our revised manuscript.

Page: 6, Lines: 111-114

We chose 0.1 (S/C) as one cutoff value because we hypothesized that, based on our experience, many non-COVID-19 patients would have antibody levels below 0.1. Other potential cutoff values were selected between 0.1 and 1.4.

In the result section:

1-4 Line115: Add the total number and percentage of COVID-19 cases in text.

RESPONSE 1-4

We thank you for your suggestion. Accordingly, we added the total number and percentage of COVID-19 cases to the results section of our revised manuscript.

Page: 7, Line 126

The total number of RT-PCR-positive patients was 56 (5.2%).

1-5 Line 126: Mention that percentage was calculated from total number of tested cases.

RESPONSE 1-5

We thank you for your suggestion. Accordingly, we added this information to the results section.

Page: 9, Line 136-137

Out of the 1,070 individuals who underwent all three tests, rapid antigen tests were positive in 42 patients (3.9%)

1-6 Line 141: you stated that only 4 tested positive on both rapid tests which doesn’t align with your data. In Line 169-175 as well as in line 210-211 you mentioned that these four cases were negative. Please correct the information.

RESPONSE 1-6

We thank you for pointing out this and apologize for this unintended mistake. We corrected it in the results section of our revised manuscript.

Page: 10, Line 153-154

Regarding the RT-PCR positive patients, only four patients tested negative on both rapid antigen and antibody testing.

In the discussion section:

1-7 Line 199-200: you could include more details about the synthesis time of each of the antibodies.

RESPONSE 1-7

We thank you for your suggestion. We added more details as suggested.

Page: 14, Lines: 213-219

Similarly, the sensitivities of IgM were 23.2% (95% CI: 14.9–34.2), 58.4% (95% CI: 45.5–70.3), and 75.4% (95% CI: 64.3–83.8) during the first, second, and third week, respectively. In addition, the sensitivities of IgA were 28.4% (95% CI: 0.9–94.3), 78.1% (95% CI: 9.5–99.2), and 98.7% (95% CI: 39.0–100) during the same three weeks, respectively. Furthermore, the sensitivities of combination of IgG/IgM were 30.1% (95% CI: 21.4–40.7), 72.2% (95% CI: 63.5–79.5), and 91.4% (95% CI: 87.0–94.4) during these same three weeks, respectively.

1-8 Line 233: state the actual accuracy of RT-PCR in this context.

RESPONSE 1-8

We thank you for your suggestion. Since there was no uniform method for retesting and follow-up after first RT-PCR, we suggested it to be an accurate explanation. Therefore, we deleted the sentence.

In the references section

1-9 There are two set of references! Please revise and combine them.

RESPONSE 1-9

We apologize for this unintended mistake. We revised them.

Reviewer #2: 

2-1 The manuscript described the performance of combined rapid antigen and antibody test. The results showed an improved diagnostic accuracy, with 92.9% sensitivity, 98.0% specificity. These data can be useful to clarify the role and the potential utility of the combination of rapid antigen and antibody tests that could be an alternative to RT-PCR testing. The manuscript is well written. 

However, the paper needs some minor revision before being published.

The authors should indicate which statistical test used when they made the comparison of the number of days after symptom onset for rapid antigen testing and IgG antibody testing. The authors report only the significances with a p equal to 0.0001. They also report the mean and standard deviation values, where sometimes the standard deviation is higher than the mean. Perhaps they need to check the distribution of the data and choose the most appropriate statistical test.

RESPONSE 2-1

We thank you for your suggestion and encouraging words. Accordingly, we changed from mean and standard deviation to median and IQR, and from t-test to Wilcoxon rank sum test. We added the explanation of statistical test to the methods section and modified the results accordingly.

Page: 7, Lines: 116-118

We obtained median (interquartile range, IQR) for continuous variables and frequency (%) for categorical variables. In addition, we used the Wilcoxon rank sum test to compare between the two groups by antigen and antibody results.

Page: 12, Lines: 175-180

The median (IQR) of the number of days from symptom onset was 5 (3-7) days in rapid antigen-positive patients and 12 (7-18) days in rapid antigen-negative patients, showing a statistically significant difference (p = 0.0009). Similarly, IgG antibody test results showed the number of days from symptom onset to be 12 (7-18) days in IgG antibody-positive patients and 4 (2-7) days in IgG antibody-negative patients, also with a statistically significant difference (p < 0.0001).

---

## [Editor Report · Decision Letter 1]

17 Jan 2022

Rapid screening for severe acute respiratory syndrome coronavirus 2 infection with a combined point-of-care antigen test and an immunoglobulin G antibody test

PONE-D-21-33926R1

Dear Dr. Mori,

We’re pleased to inform you that your manuscript has been judged scientifically suitable for publication and will be formally accepted for publication once it meets all outstanding technical requirements.

Kind regards,

Gheyath K. Nasrallah

Academic Editor

PLOS ONE
---

## [Editor Report · Acceptance letter]

24 Jan 2022

PONE-D-21-33926R1 

Rapid screening for severe acute respiratory syndrome coronavirus 2 infection with a combined point-of-care antigen test and an immunoglobulin G antibody test 

Dear Dr. Mori:

I'm pleased to inform you that your manuscript has been deemed suitable for publication in PLOS ONE. Congratulations! Your manuscript is now with our production department. 

Kind regards, 

on behalf of

Dr. Gheyath K. Nasrallah 

Academic Editor

PLOS ONE